# Improving the Chemical Properties of Acid Sulphate Soils from the Casamance River Basin

**Inmaculada Bautista [1],\*, Joana Oliver [1], Antonio Lidón [1] , Jose María Osca [2] and Neus Sanjuán [3]**

[1] Research Group in Forest Science and Technology (Re-ForeST), Research Institute of Water and Environmental Engineering (IIAMA), Universitat Politècnica de València, Camí de Vera s/n, 46020 Valencia, Spain; alidon@qim.upv.es (A.L.)

[2] Departamento de Producción Vegetal, Universitat Politècnica de València, 46020 Valencia, Spain

[3] Research Group in Analysis and Simulation of Food Processes (ASPA), Departamento de Tecnología de Alimentos, Universitat Politècnica de València, 46020 Valencia, Spain; nsanjuan@tal.upv.es

\* Correspondence: ibautista@qim.upv.es

**Abstract:** The anoxic conditions produced after the reflooding of acid sulphate soil (ASS) can reduce sulphate and/or Fe(III) with a consequent rise in pH. This study aimed to compare the effect of different amendments on ASS remediation and to analyse the effect on soil pH and exchangeable aluminium. Two mid-term incubation experiments were carried out to analyse the effect of amendments and water management on ASS. Soil samples were taken in the Santak Valley from four agricultural plots. During the first experiment, each soil sample was subject to two water management systems (flooded and non-flooded) and three amendment types (rice straw, manure, and lime). During the second experiment, the flooded condition was performed with three organic amendments (rice straw, manure, and biochar). In the first experiment, the amendments with organic matter (rice straw, and manure) increased the pH more under the flooded conditions, and manure was effective in reducing exchangeable aluminium ($Al_{ex}$) to 45% in the control soil. In the second experiment, all the organic amendments reduced soluble Al, but whereas straw increased soluble Fe, biochar diminished it. The amendment addition increased the soil pH and reduced $Al_{ex}$. The $Al_{ex}$ reduction was greater for the stabler organic amendments: manure and biochar.

**Keywords:** iron toxicity; acid sulphate soils; exchangeable aluminium; redox potential; organic amendments

## 1. Introduction

Acid sulphate soil (ASS) includes all the soils and materials in which, as a result of soil formation, sulphuric acid will be, is, or has been produced in amounts that have a lasting effect on the main soil characteristics [1]. The bulk of ASS has been geographically associated with modern coastlines worldwide. Sulphidic material naturally formed in coastal areas during the elevation of sea level 10,000 years ago, when seawater rich in sulphate ions was reduced under anoxic conditions by heterotrophic and chemoautotrophic bacteria, especially in soils with high organic matter content [2,3]. Of the estimated 17–14 Mha of ASS worldwide, 4.5 Mha are in Africa [2].

Acid soils with sulphidic material are productive under waterlogged conditions, constituting one of the most productive wetlands. With drainage, however, the oxidation of sulphidic material forms sulphuric acid, which lowers the soil pH and makes soil unproductive for most uses. Under oxidative conditions, the first pyrite oxidation stage results in ferrous iron, acid, and sulphate formation. Ferrous iron is further oxidised to ferric iron by a much faster reaction catalysed by *Acidothiobacilus ferrooxidans* at a low pH [4]:

$$FeS_{2\,(s)} + 7/2\,O_{2\,(aq)} + H_2O \rightarrow Fe^{2+}_{\,(aq)} + 2\,H^+_{\,(aq)} + 2\,SO_4^{\,2-}_{\,(aq)}$$

$$Fe^{2+}_{\,(aq)} + H^+_{\,(aq)} + \frac{1}{4}\,O_2 \rightarrow Fe^{3+} + \frac{1}{2}\,H_2O$$

Low pH produces the solubilisation of toxic elements, such as Al and Mn, which inhibit root growth. Aluminium toxicity is a critical growth-limiting factor for plants in many acid soils, mainly affecting root growth, especially when soil pH values are less than 5.0 [5]. The toxicity to the plant roots is determined by the availability of monomeric species of Al [6,7]. Phytotoxicity loss occurs when monomeric Al is diminished by the polymerization of Al by increasing pH [7,8], forming complex non-exchangeable polymeric hydroxy-aluminum ions [9]. Dent [10] recommended keeping ASS flooded to prevent oxidation or carrying out activities that cause a minimum disturbance, such as flooded rice cultivation or making shallow ponds for aquaculture.

Rice is estimated to be grown on 4 million hectares (Mha) of ASS, mainly in the Mekong Delta and coastal areas of Africa (Senegal, Gambia, Guinea Bissau, Sierra Leone, Liberia), SE Asia (Indonesia, Thailand) and South America (Venezuela, the Guyanas) [11]. The most critical rice-growing constraints in ASS are aluminium toxicity and phosphorous deficiency, which are associated with low pH levels and Fe toxicity in field trials [12,13].

In the tropics, the distinct dry-wet seasons form a seasonal pattern, with more acid produced during the dry season [1,14]. During the wet season, after the resaturation of sulphuric soils and the consequent lowering of the redox potential, the reduction brought about by reducing bacteria can lead to a rise in pH [15]. This increase in pH after flooding is associated with a steep rise in dissolved $Fe^{2+}$ in young ASS, which still contains large amounts of partially decomposed organic matter.

Therefore, seasonal flooding in ASS triggers chemical and biological processes. Specifically in flooded rice fields, oxygen is depleted, and ions like $NO_3^-$, $Mn^{4+}$, $Fe^{3+}$, and $SO_4^{2-}$ act as electron acceptors for microbial respiration and sequentially become reduced, which results in a large amount of soluble Fe that, coupled with the water flow from uplands to lowlands, results in toxic dissolved iron levels that can diminish rice yields by 12–100% [16]. In highlands and on slopes, iron remains in more insoluble ferric forms. Conversely, under more anoxic conditions in lowlands, iron is reduced to more soluble and mobile ferrous forms visible in rusty soil or as oily stains in water [17]. The amount of extractable $Fe^{2+}$ is enhanced by a low initial soil pH, sustained organic matter supply [18], and the absence of compounds in a higher oxidation state than Fe (III) oxide [19]. Soil type, organic matter content, and fertility, together with cultivars and microbial activities, determine the magnitude of these chemical changes [20].

A high iron concentration produces iron toxicity in rice, a disorder syndrome associated with high iron concentrations in a soil solution. Although most mineral soils are rich in iron, the expression of toxicity symptoms in leaf tissues and smaller rice yields occur only under specific flooded conditions, which involve the microbial reduction of insoluble Fe (III) into soluble Fe (II) [20]. Iron toxicity only occurs in flooded soils, and hence, it primarily affects lowland rice production [21]. This nutritional disorder has also been associated with other nutrient deficiencies, such as phosphorous, potassium, and zinc deficiencies [22].

Rice is a staple food in the Senegalese diet. Senegal is the third-largest rice importer in Africa after Nigeria and South Africa [23]. Rice consumption represents around 31% of the calorie intake in Senegal, whose domestic rice production only covers about 28% of the country's requirements [24]. Rice is primarily cultivated in the Senegal River valley, where it is mostly irrigated, and in the Casamance River basin as a rainfed crop. Iron toxicity is frequent in the lower Casamance basin, unlike the other basin areas (the Senegal, Gambia, and Saloum Rivers) [22].

In the Casamance region, the climate is characterised by two differentiated seasons: dry and wet. In this area, rice production is rainfed, with yields varying from 1 to 2 Mg·ha$^{-1}$. The causes of such low yields are low applied inputs and the use of low-yielding traditional varieties with medium- or long-duration cycles (Caritas Ziguinchor, personal communication, 2018). The coastal west Senegal region is lowland a few metres above sea level. Rice cultivation is traditionally widespread and characterised by the following peculiarities: (i) it is essentially carried out by women; (ii) almost all rice-growing operations are per-

formed manually with traditional tools; (iii) fertilisers and pesticides are not or barely used; (iv) it is carried out on very small plots [25].

A historical analysis of rainfall patterns revealed a decreasing trend in annual rainfall in Southern Senegal for the 1922–2015 period [26], which evidences that this region will become drier. The Casamance estuary presents a point of maximum salinity migrating from 20–30 km upstream by the end of the dry season to 50–80 km downstream from this point at the end of the rainy season [27]. A dry period between 1970 and 1992 caused severe soil degradation from secondary valleys by acidification and salinisation [28]. During the years 1980–1985, small dams were built at the bottom of secondary valleys to prevent salinisation [29,30]. The analysis of flood and drought episodes suggests the need to improve water management systems to reduce the risks of yield loss [31]. The local topography is relatively flat, making building reservoirs for water management difficult. The consequences of rainfall shortages causing salinisation have contributed significantly to agriculture gradually being abandoned. The quantitative results of the land-use dynamics based on satellite data showed a significant decline in the rice area for 1984–2010 in the subwatersheds of the lower Casamance basin. However, increased rainfall reversed this trend between 2000 and 2010 [32].

The abovementioned situation highlights the need for adaptation strategies to maintain rice productivity in the Casamance Valley. According to Becker and Asch [21], the management strategies to help face high iron concentrations and improve rice yields include the selection of tolerant rice varieties that are associated with better soil management practices, such as water management, enhanced drainage, and applying nutrients with positive effects on plant response (P, Zn, or K).

Two kinds of strategies are applied to deal with ASS [33]: (a) using alkaline amendments to neutralise acidity; (b) preventing the oxidation of sulphidic material. Diallo [34] performed a 2-year experiment under field conditions to analyse the crop response to soil amendments, namely lime, oyster shell, and biochar. Only lime and oyster shell increased pH by conferring higher-yield parameters. Ebimol et al. [35] recommend applying phosphogypsum and lime to increase available Ca and soil pH and lower toxic iron and aluminium levels. Devi et al. [36] found that applying lime or dolomite as liming material significantly improved rice yields, mainly when applied in two splits as basal and 30 days after sowing. Nevertheless, lime is expensive and unavailable to farming communities in many countries. Another feasible option is to apply organic matter, which has been used in agriculture for millennia to improve soil fertility and recycle nutrients. Several studies have reported increases in pH after adding plant organic matter [37]. However, the quality of organic matter conditioned the response to the amendment, which was more effective if incorporated into the soil or when it had a high nitrogen content [37]. Compared with complex organic matter, adding single compounds, such as glucose or nitrogen salts, resulted in a slight pH increase [33]. Adding organic amendments has also been demonstrated to increase rice production, although its efficacy also depends on the type of organic amendment applied. Halim et al. [38] concluded that adding compost as a soil amendment could increase soil pH and create favourable soil conditions for rice cultivation in ASS, improving rice performance. Ferdous et al. [39] found that applying lime and farmyard manure increased rice yields. Ghosh et al. [40] observed that adding farm manure or vermicompost rather than straw increased production. Masulili et al. [41] found that the most effective amendment to ASS in reducing exchangeable Al and Fe was the application of rice husk biochar, which also increased the rice dry matter 2.6-fold versus the control.

This study aimed to compare the effect of different amendments on ASS remediation and to (a) determine the kinetic effects of adding amendments on soil solution composition; (b) analyse its effects on soil solution pH; and (c) analyse the effect on soil pH and exchangeable aluminium. Our hypothesis was that adding organic matter would reduce the soil solution redox potential because the activity of anaerobic microorganisms would increase, and both sulphate and Fe (III) would reduce. Fe (III) reduction would increase pH by

affecting iron solubility. The different amendments were used to test the hypothesis about the effect of organic matter composition on microbial activity and acidity neutralisation.

## 2. Materials and Methods

### 2.1. Research Area and Soil Sampling

The climate is tropical in the low Casamance region, and rainfall is concentrated mainly from July to September. According to the Köppen–Geiger classification, the climate is Aw, with an average annual temperature of 26.7 °C, and the average annual rainfall is 1269 mm. The driest month is January, with 0 mm, whereas August is the wettest, with a mean rainfall of 429 mm. The soil from the Casamance basin presents low K, Mg, Ca, and Na values and good total P and total N levels [42].

Rainfed rice cultivation has been traditionally carried out in the lowlands of the Casamance River basin (South Senegal). These areas are influenced by seawater salinity, with tidal flows as far as 100 km inland from the coastal line [43]. In one of the secondary valleys, the Santack Valley near Ziguinchor (Figure 1), Caritas Spain collaborated with Caritas Ziguinchor and funded a project to recover soil from growing rainfed rice. In the two first campaigns carried out during the 2017 and 2018 wet seasons, new rice varieties (NERICA, Sahel 108 and Sahel 201) were compared with traditional ones. In general, local varieties better tolerate iron toxicity but have a longer growing period (4 to 5 months). Rice yields range from 0 in the lower part of the valley to 2 Mg ha$^{-1}$ in the upper part. This study is framed within a 2-year research project (AD1810-UPV) carried out jointly with Caritas Spain and Caritas Senegal in 2019–2020, with funds from the Centre for Development Cooperation of the Polytechnic University of Valencia (Universitat Politècnica de València, Valencia, Spain).

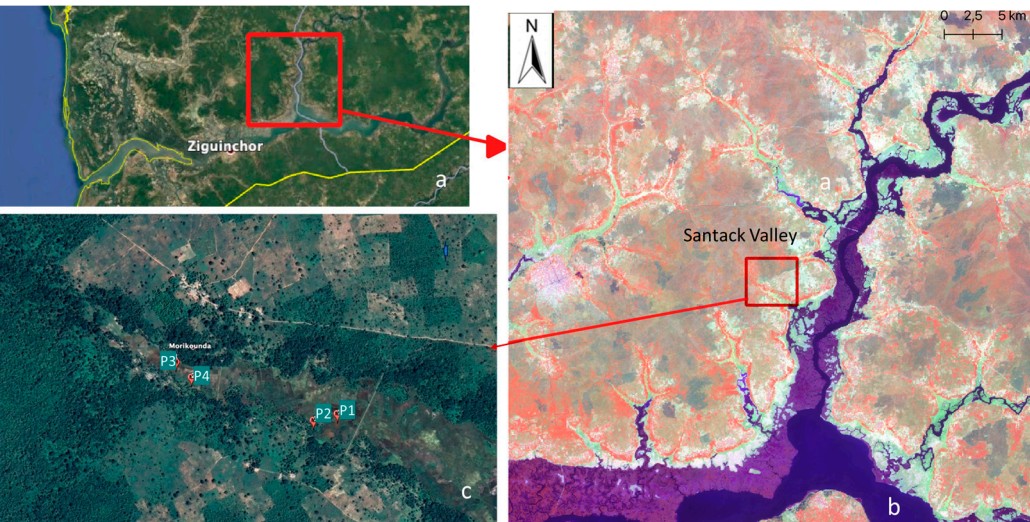

**Figure 1.** Map of the Casamance River Basin (**a**); location of the Santack Valley (false color) (**b**); and P1, P2, P3 and P4 plot locations in the Santack Valley (**c**).

In July 2019, four agricultural plots were selected, as advised by the technique staff from Caritas Ziguinchor. Plots (labelled P1, P2, P3, and P4) were selected for their productivity differences in the previous campaign. Specifically, plot P1 had null productivity in the last campaign, and the remaining plots located in the highest level of the valley were more productive. To characterise soil properties, a composite soil sample was obtained in each plot, made of 4 samples taken from 0 to 10 cm. A single soil sample was taken in the plateau near the valley from 0 to 10 cm. Soil samples were transported to the laboratory, air-dried, ground, and sieved through a 2 mm mesh. Electrical conductivity (EC) and pH were measured in a 1:2.5 (*w/v*) aqueous solution with a conductivimeter (Crison GLP31, Vizcaya, Spain) and a pH meter (Crison micropH 2000, Vizcaya, Spain), respectively. Soil texture was determined by the Bouyoucos method, according to the USDA limits: sand (2–0.05 mm); silt

(0.05–0.002 mm); and clay (less than 0.002 mm). Soil organic carbon (SOC) was determined by wet oxidation with 1 N potassium dichromate in an acidic medium and by evaluating excess dichromate with 0.5 N ferrous ammonium sulphate, as described by Walkley and Black [44]. The soluble sulphate content was determined by turbidimetry in a 1:5 soil water extract [45]. Exchangeable aluminium ($Al_{ex}$) was extracted with KCl 1.0 mol $L^{-1}$ with a 1:10 soil: solution ratio and shaking for 10 min. Exchange acidity was determined using a bromothymol blue indicator by titration with a standardised 0.025 mol $L^{-1}$ NaOH solution. Exchangeable Al was obtained by subsequent titration with 0.025 mol $L^{-1}$ HCl after the complexation of Al with NaF 1 M [46,47].

All the soils were fine textured, with a high silt and clay content (meanly silty clay or silty clay loam textural class) and very acidic (pH around 4) with low organic matter content (Table 1). The soil from the plateau high up in the valley had a lower clay content and a higher pH. Regarding salinity, P1 had a high soluble salt content (both the salinity and the sulphate content were higher). This plot, located in the depressed area of the valley, received drainage water from the upper plots. The soils from the four plots had a very low pH, with plots P1 and P3 showing the lowest values. The soil organic matter levels were quite low, especially on the plateau. Oxidable organic carbon was around 6 g/kg dry weight on the surface soil.

**Table 1.** Soil characterisation.

| Plot | pH 1.2,5 Soil/Water | Sand | Silt | Clay | EC 1:5 Extract | Soluble $SO_4^{2-}$ | Oxidable Organic Carbon | $Al_{ex}$ |
|---|---|---|---|---|---|---|---|---|
| | | % | % | % | dS/m | mg/kg | g/kg | cmolc/kg |
| P1 | 4.01 | 17.5 | 42.6 | 39.9 | 0.73 | 689 | 6.20 | 4.76 |
| P2 | 4.16 | 14.4 | 43.3 | 42.3 | 0.33 | 146 | 6.27 | 3.54 |
| P3 | 4.02 | 11.9 | 53.5 | 34.5 | 0.35 | 155 | 3.83 | 3.27 |
| P4 | 4.26 | 15.7 | 37.9 | 46.4 | 0.17 | 54 | 5.71 | 3.43 |
| Plateau | 4.43 | 28.4 | 34.6 | 37 | 0.28 | 231 | 0.5 | 1.27 |

Additionally, at one point in each plot, two samples were taken at different depths (topsoil 0–10 cm and subsoil 10–20 cm), except for plot P3, for which only the subsoil sample was available, and plot P4, where only the topsoil sample was taken. These soil samples were transported to the laboratory at field humidity, gently sieved through a 2 mm mesh, and left at field humidity at 4 °C until the incubation experiments were performed.

### 2.2. First Incubation Experiment

The soils selected for the first incubation experiment corresponded to the surface samples, except for plot P3, where the subsoil layer was used. A factorial design was applied by considering two factors of variation: (a) water management with two levels (flooded vs. free drainage); (b) chemical amendment with four treatments (control soil, soil + rice straw, soil + manure and soil + lime).

Rice straw was obtained from a rice production zone in L'Albufera (Valencia, Spain) in September 2019. The sample was air-dried, ground, and sieved through a 0.5 mm mesh size. The manure was sheep manure obtained from an organic agricultural plot in Pedralba (Valencia, Spain). The manure was air-dried, ground, and sieved through a 0.5 mm mesh. Total C and total N were determined in the soil samples and organic amendments using a total analyser (FLASH EA 1112 SERIES-LECO TRUSPEC). P, K, Ca, S, Al, Fe, and Mg contents were determined by inductively coupled plasma optical emission spectroscopy (ICP-OES; ICAP 6500 DUO/IRIS INTREPID II XDL) after acid digestion ($HNO_3^-H_2O_2$ 4:1) in a microwave.

Then, 0.5 g of the amendment was added to 40 g of soil; that is, 12.5 mg amendment $g^{-1}$ soil. 4.8 and 2.8 g C $kg^{-1}$ soil were added with straw and manure organic amendment,

respectively, which was similar to the values of oxidable organic carbon initially found in the soils.

Each amended soil sample was placed into a lockable percolation tube and saturated with 25 mL of distilled water. The percolation tubes were hand-made of glass; thus, they were not uniform. The internal diameter and height were around 3.4 and 12 cm, respectively. The soil depth inside the percolation tube was around 5 cm. In the tubes corresponding to the flooded treatments, where drainage was not allowed, 40 mL of distilled water was added. In this way, in the flooded treatments, the soils had a constant water sheet around 4.5 cm in height. In the free drainage treatment, the water exit was left open; in this way, the soils were kept at field capacity humidity. A total of 32 incubation tubes were prepared (4 soils × 4 amendment treatments × 2 water management treatments). Soils were incubated for 14 weeks at room temperature (around 20 °C), and the soil solution was extracted regularly by percolation at set times: 5, 12, 26, 40, 68, and 96 days from the time incubation began. Soil solution was recovered from the flooded treatment by allowing drainage and, after extracting soil solution, the exit was locked, and 40 mL of distilled water was added. In the non-flooded treatments, the drainage water was collected after adding 40 mL of distilled water. The pH and redox potential of percolates were measured by electrochemical methods: combined pH electrode and Pt-electrode. Soluble Fe was determined by atomic absorption spectrometry (Varian). Soluble sulphate was determined by turbidimetry at 450 nm [45]. After 2 weeks, some percolates were brown in colour. For this reason, absorbance at 450 nm was also measured in solution and used as a blank to correct the sulphate determinations.

At the end of the experiment, soils were dried and sieved through a 2 mm mesh. Exchangeable Al ($Al_{ex}$) was extracted with KCl 1.0 mol·L$^{-1}$ solution in a 1:10 soil: solution and determined as previously described [46,47]. The final soil pH was measured in a 1:2.5 (*w/v*) aqueous solution by a pH meter (Crison micropH 2000, Spain).

### 2.3. Second Incubation Experiment

During the second experiment, soil samples were incubated at 25 °C in the dark in hermetically sealed flasks with septum stoppers, and only the flooded treatment was considered. Four amendment treatments (control soil, soil + rice straw, soil + manure, and soil + biochar) were performed for all the sampled soils. The incubations were performed for each combination of soil and treatment, considering the soils as replications of the amendment treatments. The biochar was purchased from Piroeco Bioenergy S.L. (Malaga, Spain). It was produced from holm oak by slow pyrolysis at 650 °C and atmospheric pressure, and the residence time in the reactor chamber was 12–18 h. Biochar was air-dried, ground, and sieved through a 0.5 mm mesh. The elemental composition of the biochar was obtained from Saez et al. [48]. The quantity of the amendment was the same as that for the first incubation, i.e., 0.5 g of amendment to 40 g of soil. The corresponding organic carbon added with biochar was 7.8 mg C kg$^{-1}$ soil. To maintain the flooded condition, 70 mL of distilled water was added to each flask.

The respiration rate was determined periodically for short times (first 4 h and the first day), twice a week for the first 4 weeks, and weekly until 6 weeks. It was calculated from the % $CO_2$ increment in the headspace volume of the flask, which was measured with a $CO_2$ sensor (Checkpoint, PBI Dansensor, Ringsted, Denmark) and expressed per mass of dry soil and time. After measuring the respiration rate, septa were removed, and the soil suspension was agitated for 1 min to measure pH and the redox potential with a potentiometer (ORP, PCE-228, PCE Ibérica, Spain). Septa were used again to isolate samples to continue with incubation. At the end of the incubation, 15 mL of the supernatant suspension was filtered through 45 μm Millipore filter to analyse the elemental composition by ICP-OES (ICAP 6500 DUO/IRIS INTREPID II XDL). The soil was dried and sieved through a 2 mm mesh. $Al_{ex}$ was extracted with KCl 1.0 mol L$^{-1}$ solution with a soil: solution ratio of 1:10 for a 10-min shaking time and determined as previously described.

*2.4. Statistical Analyses*

The differences in the solute concentration of soil leachates during every irrigation treatment due to each amendment addition were determined by one-way analysis of variance (ANOVA), followed by Tukey's multiple range test ($p < 0.05$). The differences between irrigation treatments (flooded vs. free drainage) were assessed by a paired t-test. Two-way ANOVA was used to detect the interaction effects of the amendment addition and irrigation treatment on the leachate chemical composition for each extraction time. All the statistical analyses were performed with the Stratigraphic Centurion XVIII software package for Windows (Statpoint Technologies, Inc., Warrenton, VA, USA).

## 3. Results

*3.1. Soils and Amendments Elemental Composition*

The soils used in the incubation experiments had different elemental compositions (Table 2), and the subsurface horizon of plot P3 had the lowest nutrient content. Once again, the samples from plot P1 had higher S and Fe values, which indicated an exogenous input of these elements by percolating water. The oxidable organic carbon obtained with the Walkey–Black wet digestion method was around 20% of the total carbon values obtained by dry combustion, whereas it is normally considered that carbon recovery by the wet digestion method is around 77% [44]. Ghosh et al. [40] also found a low oxidable organic carbon recovery rate with the Walkey–Black method in other acid soils.

**Table 2.** Elemental composition (g kg$^{-1}$) in the material used during the incubation experiments.

| Soil | Depth | C | N | C/N | P | S | K | Ca | Mg | Fe | Al |
|------|-------|-----|------|------|------|-----|------|------|------|------|------|
| P1 | (0–10) | 41.1 | 3.7 | 11.0 | 0.49 | 2.5 | 1.28 | 0.3 | 0.79 | 42.1 | 62.7 |
|  | (10–20) | 22.6 | 2.0 | 11.3 | 0.21 | 0.9 | 1.39 | 0.1 | 0.84 | 13.8 | 90.8 |
| P2 | (0–10) | 30.8 | 2.5 | 12.0 | 0.35 | 0.8 | 1.24 | 0.2 | 0.67 | 16.1 | 52.2 |
|  | (10–20) | 25.7 | 2.1 | 12.2 | 0.24 | 0.5 | 1.05 | 0.1 | 0.54 | 9.8 | 43.4 |
| P3 | (10–20) | 17.4 | 1.6 | 11.0 | 0.15 | 0.6 | 1.00 | 0.1 | 0.57 | 7.5 | 57.3 |
| P4 | (0–10) | 28.2 | 2.3 | 12.2 | 0.32 | 0.6 | 1.22 | 0.1 | 0.70 | 21.9 | 63.5 |
|  | Rice straw | 387.4 | 10.5 | 36.7 | 0.72 | 1.3 | 11.7 | 3.1 | 1.73 | 0.2 | 0.2 |
|  | Manure | 220.8 | 19.2 | 11.5 | 7.09 | 6.5 | 11.9 | 37.0 | 70.6 | 7.3 | 4.4 |
|  | Biochar * | 627 | 9 | 70.0 | 1.80 | 0.1 | 7.7 | 37.6 | 4.0 | 1.6 | nd |

* Data from [48]. Nd, non-determined.

The selected plots differed in productivity terms; the P1 plot, located near the middle slope dam, had the lowest yield. The soil characteristics (Table 1) showed that this low productivity was associated with both high salinity and soil sulphate content because the soil pH was similar in all the plots. The high salinity level revealed a secondary salinisation process worn by the superficial drainage water from the upper plots, which transported soluble oxidation products like $SO_4^{2-}$ but also soluble metals, as shown by the higher $Al_{ex}$ content of this soil.

*3.2. First Incubation Experiment*

3.2.1. Evolution of pH in Percolates

Figure 2 clearly shows that during incubation, both water management and amendment addition influenced the pH of the percolates. During the incubation experiment with the flooded soils (Figure 2a), the addition of organic amendments increased the drainage pH by 0.5 units in the first 5 days and by around 1.5 units in the following 7 days. Lime addition increased the pH values to more than 7 in the first 12 days, and this value remained throughout the incubation experiment. Some gas emanation was visually observed, with bubbles forming inside the soil with the lime treatment. The emitted gas was probably $CO_2$ produced by neutralising the acidity with lime. In the control soils, the pH gradually rose to

values of around 6 after 40 days of incubation. In the free drainage treatment, only the lime treatment significantly increased the soil pH in relation to the control. At the end of the experiment, the organic amendments had significantly increased the pH of the percolates.

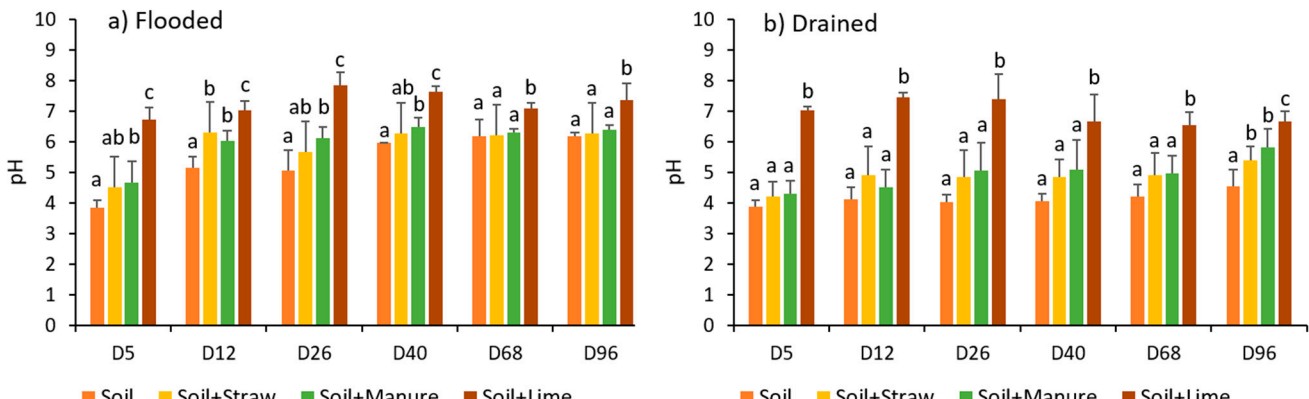

**Figure 2.** pH evolution in the percolates on days 5, 12, 26, 40, 68, and 96 after incubation started: (**a**) flooded soils; (**b**) drained soils. Results shown correspond to the mean and standard deviations of four samples ($n = 4$). Statistically significant differences at $p \leq 0.05$ due to amendments are indicated with lowercase letters.

### 3.2.2. Evolution of the Redox Potential in Percolates

The redox potential was controlled by the secondary redox reactions associated with amendment and oxygen depletion under waterlogged conditions. Figure 3 exhibits the evolution of the redox potential of percolates during the first incubation experiment. The redox potential of percolates was lower in the flooded treatment compared with the drained one, which indicates anoxic conditions (Figure 3a). The percolates obtained on day 68 indicated an anomalous trend, with a generalised increase in the potential redox. The redox potential of water extracts was normally measured on the same day, but on day 68, the redox potential measurement was delayed several days after extraction. This value could be due to the oxidation of aqueous extracts within the interval between soil extraction and the redox measurements, although samples were maintained at 4 °C to minimise biological activity. The lowest redox potential value was around 50 mV and corresponded to the treatment with straw incorporation 12 days after flooding. In general, the samples treated with lime obtained the lowest redox potential values under flooding, which indicates that pH values close to neutrality are optimal for biological activity. According to Yuan et al. [49], optimal pH values for sulphate reducers have to exceed 5. After 1 week under the flooded conditions, the water percolates from those samples treated with straw were yellow-brown in colour with a film of floating material, possibly due to the presence of Fe (III) oxyhydroxides. A fine orange layer appeared in the soil several mm under the surface after 2 weeks (Figure 4).

### 3.2.3. Evolution of Soluble Iron in Percolates

Two days after incubation began, the soluble iron content in the drained soils was high, but this content decreased after 2 weeks (Supplementary Material Table S1). As expected, the lime amendment lowered the pH to values close to neutrality, which precipitated all the iron forms. Under the flooded conditions, the organic amendments maintained the highest soluble iron levels throughout the experiment (Figure 5).

Two weeks after flooding, the percolates in the flooded treatment were brown in colour, and an oily layer floated on the water, especially in the samples amended with straw. Colour was measured at 450 nm absorbance (Figure 6). After the first 4 weeks of incubation, the values in the saturated soils amended with organic matter (straw, manure) were higher.

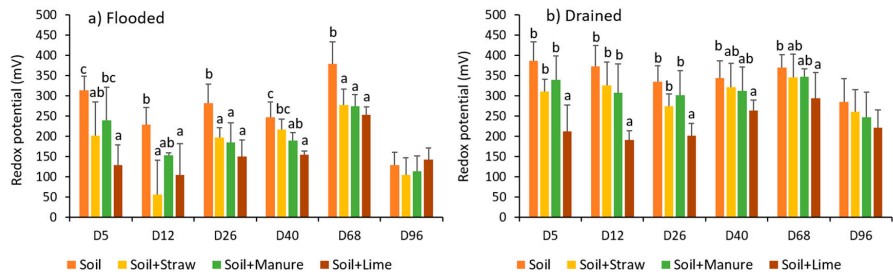

**Figure 3.** Evolution of the percolates redox potential on days 5, 12, 26, 40, 68, and 96 after incubation started: (**a**) flooded soils; (**b**) drained soils. Results correspond to the mean and standard deviation of four samples (*n* = 4). Statistically significant differences at $p \leq 0.05$ due to amendments are indicated with lowercase letters.

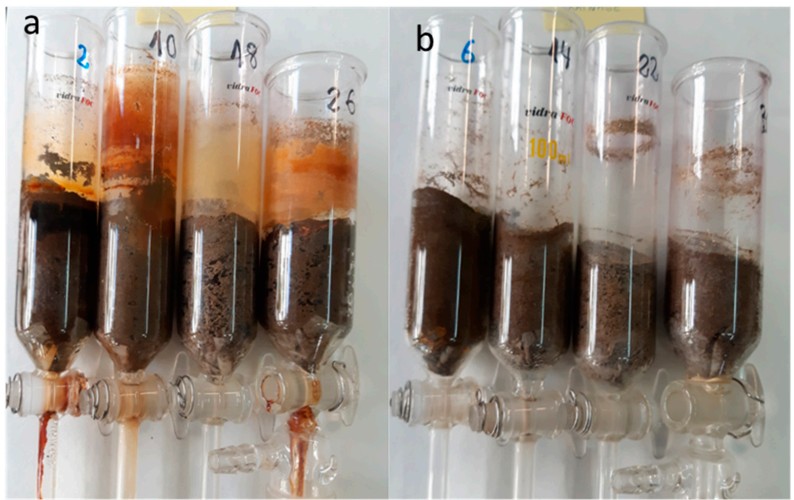

**Figure 4.** Percolation tubes of the soils amended with straw at the end of the incubation experiment in (**a**) flooded treatment; (**b**) non-flooded treatment.

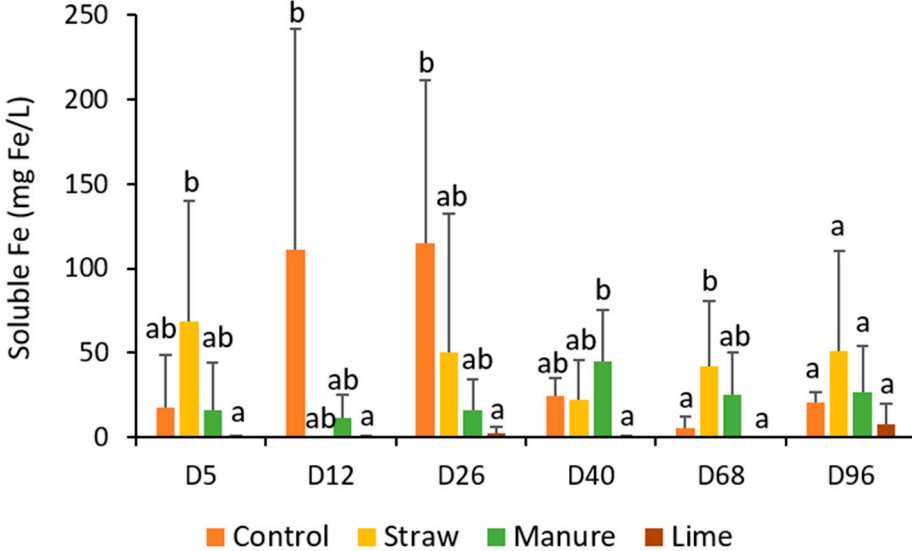

**Figure 5.** Soluble Fe evolution in the flooded treatment percolates on days 5, 12, 26, 40, 68, and 96 after first incubation experiment started. Results correspond to the mean and standard deviation of four samples (*n* = 4). Statistically significant differences at $p \leq 0.05$ due to amendments are indicated with lowercase letters.

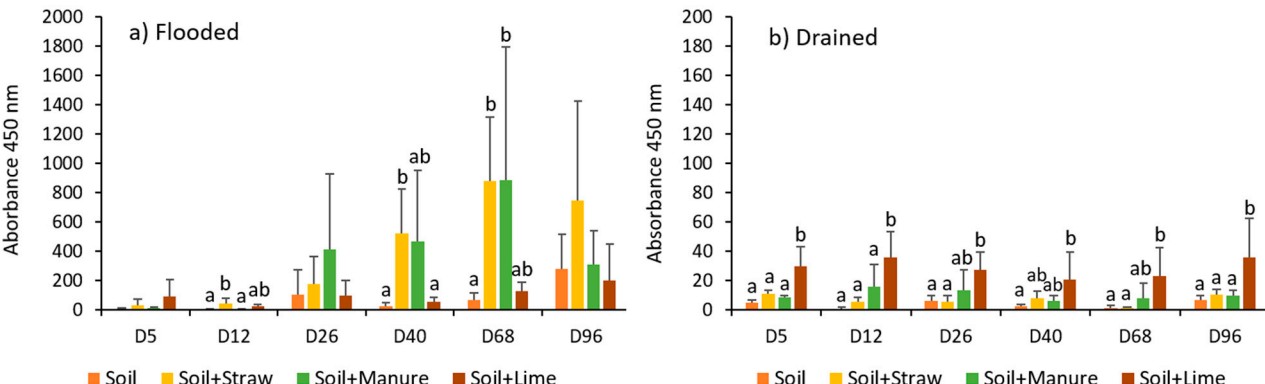

**Figure 6.** Evolution of the absorbance at 450 nm of the percolates on days 5, 12, 26, 40, 68, and 96 after incubation started: (**a**) flooded soils; (**b**) drained soils. Results correspond to the mean and standard deviation of four samples (*n* = 4). Statistically significant differences at $p \leq 0.05$ due to amendments are indicated with lowercase letters.

### 3.2.4. Sulphate Leaching

Unexpectedly, the addition of organic matter did not significantly reduce the sulphates in the soil solution (Supplementary Material Table S1). As soils differ in terms of salinity and initial soluble sulphate content, the sulphate that leached from the soil presented wide variability, and non-significant differences among treatments were statistically detected. The soluble sulphate content in the soil decreased during successive leachates. Sulphate leaching showed a similar order of magnitude in the drained and flooded soils, which was fivefold higher in plot P1, which had a higher initial soluble sulphate content. In general, sulphate losses increased with lime addition, probably due to the ion $Ca^{2+}$ released during the reaction with lime in the soils, which favours losses of this anion. The production of volatile sulphur-derived compounds was detected by odour, especially in the flooded straw treatments, which indicates sulphate reduction processes.

### 3.2.5. Exchangeable Aluminium

Both amendment addition and the irrigation method determined the final $Al_{ex}$ values in the soils (Figure 7). $Al_{ex}$ was generally lower under the flooded conditions. Regarding the amendment effects, lime was the most efficient method for reducing $Al_{ex}$ due to the increase in the soil pH. Adding manure significantly lowered $Al_{ex}$ compared with the control soil under both the flooded and non-flooded conditions. The addition of fresh straw diminished $Al_{ex}$, but differences were not statistically significant compared with the control soil. Lime was the amendment with the lowest $Al_{ex}$ values ($0.29 \pm 0.21$ and $0.31 \pm 0.19$ cmol$_c$ kg$^{-1}$ in the flooded and non-flooded treatments, respectively) and the highest soil pH values ($7.14 \pm 0.37$ and $6.83 \pm 0.72$ in the flooded and non-flooded treatments, respectively). The second-best treatment for reducing exchangeable aluminium was manure, with final soil pH values of $4.54 \pm 0.34$ and $4.52 \pm 0.31$ in the flooded and non-flooded treatments, respectively. Adding straw reduced $Al_{ex}$ to 68% and 84% of the control soil values in the flooded and non-flooded treatments, respectively. Manure addition reduced $Al_{ex}$ to 45% and 44% of the control soil values in the flooded and non-flooded treatments, respectively. $Al_{ex}$ values were lower under flooded conditions. When the final soil pH was compared with the final $Al_{ex}$ content in the soil (Figure 8), the data showed a clear relationship between both variables, which indicates (a) the effect of organic matter addition on the $Al_{ex}$ level and (b) that the secondary reduction reactions produced by amendment under the flooded conditions affected aluminium solubility.

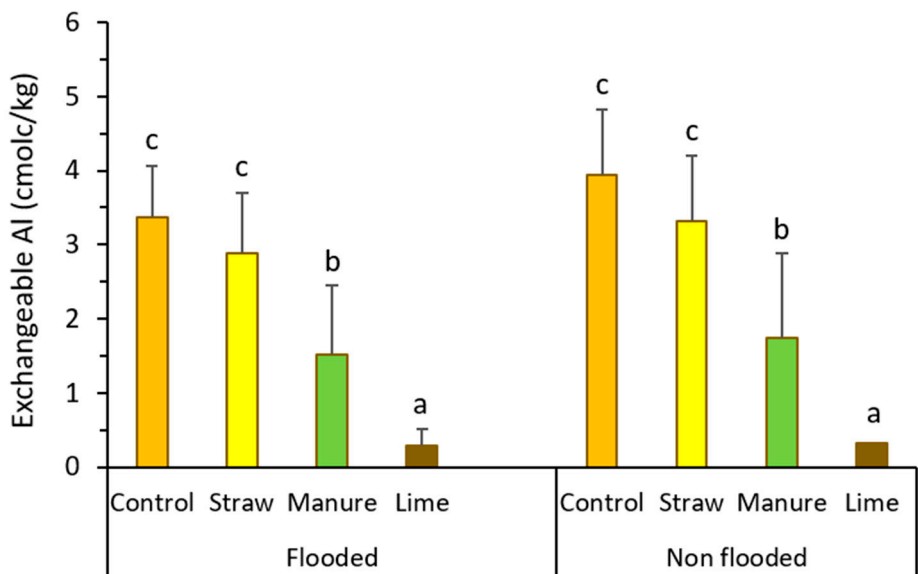

**Figure 7.** Exchangeable aluminium in soils after the first incubation experiment. Results correspond to the mean and standard deviation of four samples (*n* = 4). Statistically significant differences at $p \leq 0.05$ due to amendments are indicated with lowercase letters.

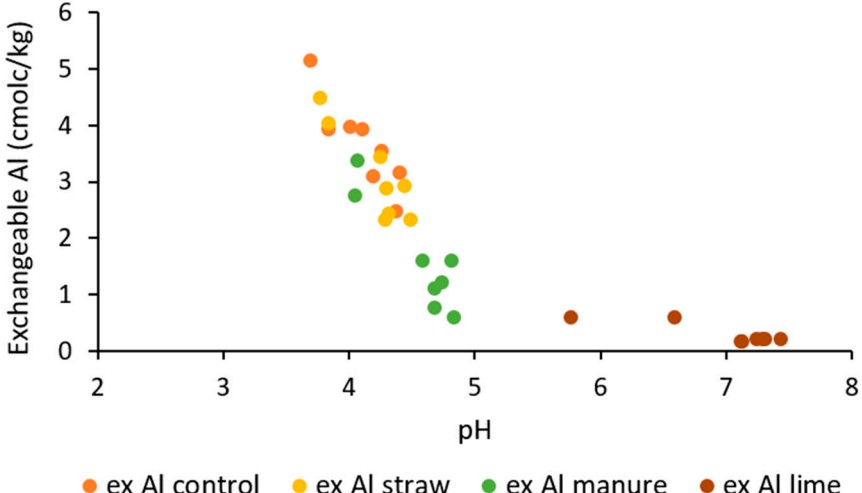

**Figure 8.** Relationship between soil pH and $Al_{ex}$ in soil after the first incubation experiment.

### 3.3. Experiment 2

In order to distinguish between chemical changes and changes due to biological activity, this second incubation was performed for the shortest time. The evolution of the biological activity, pH, and redox potential of the soil suspensions during experiment 2 is shown in Figure 9. As for $CO_2$ emissions, both biochar and manure produced high emissions for the first 4 h after adding the amendment (Figure 9a). Then, emissions gradually reduced until a constant value was obtained 12 days after incubation began, whereas the control treatment remained almost constant. In the straw treatment, $CO_2$ emissions were low at the beginning but increased between days 2 and 5, with higher emission values than those of the other treatments. For the same time interval, the straw treatment increased the pH by more than 0.5 units (Figure 9b), and the redox potential values drastically dropped to 63.83 ± 99.45 mV (Figure 9c), at which sulphate could be reduced. Sulphate reduction was also revealed by the characteristic odour of the soils amended with straw due to the production of organo-sulphur volatiles. Another characteristic that appeared only in the

straw treatment was solution turbidity, which indicates changes in the redox state of Fe accompanied by some precipitation of iron minerals.

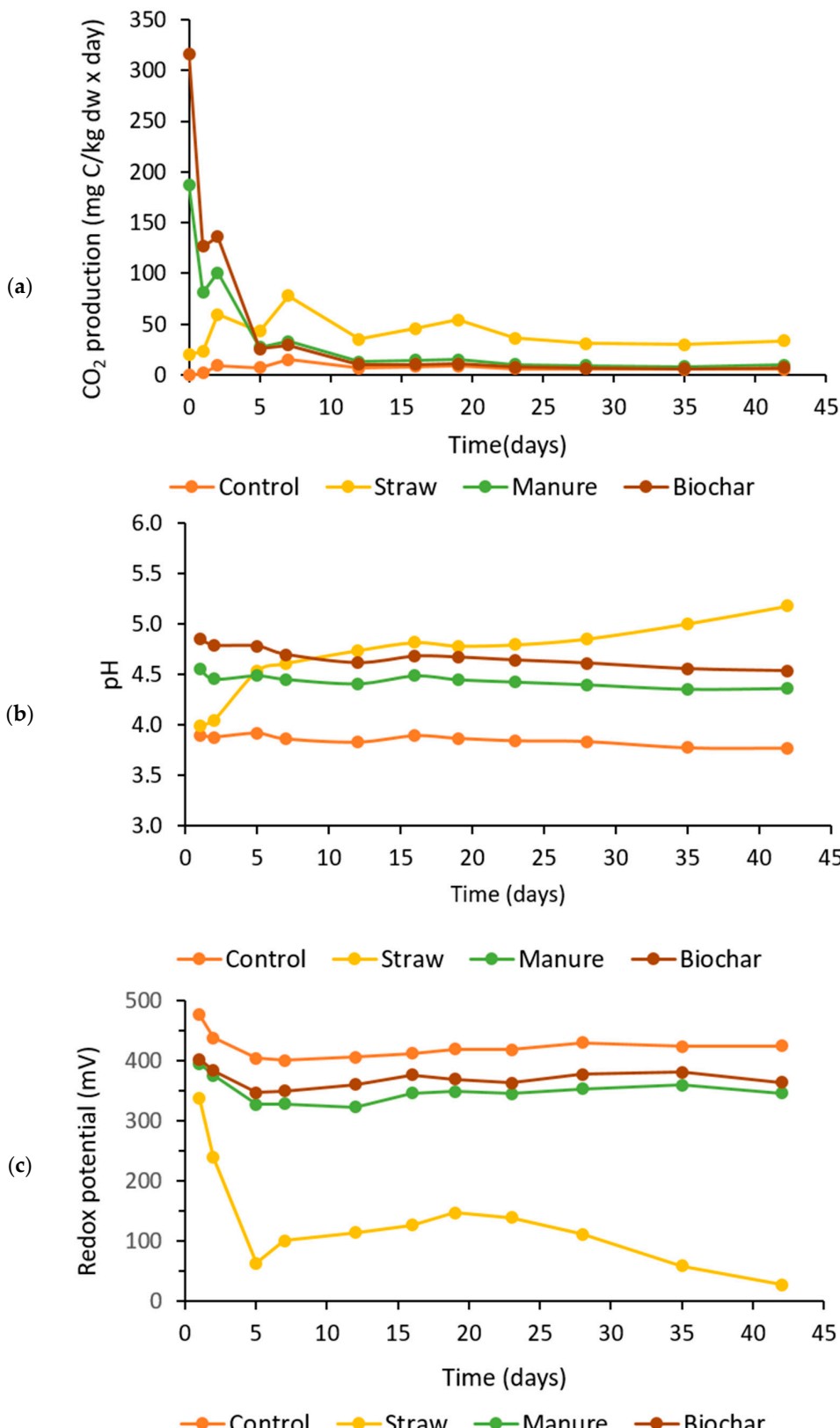

**Figure 9.** Evolution of $CO_2$ production, pH, and the redox potential with incubation time during the second incubation experiment. Data are mean values of *n* = 6.

The addition of biochar and, to a lesser extent, the addition of manure, instantly increased the suspension pH in relation to the control soil (by around 1 unit with biochar and 0.5 units with manure). These increments in the control were maintained throughout incubation.

Exchangeable aluminium was lower than 4 cmol$_c$ kg$^{-1}$ (Figure 10). This was similar to the values obtained under flooding during the first incubation experiment and implies that anoxic conditions per se slightly reduce exchangeable aluminium. In this experiment, straw addition reduced Al$_{ex}$ to 73% with respect to the control soil values, whereas the reductions obtained with manure and biochar were 34% and 27%, respectively. It must be noted that two soil samples, namely those corresponding to the samples of the P1 plot with null rice production, had a considerably higher Al$_{ex}$ for the manure and biochar treatments (Figure 11).

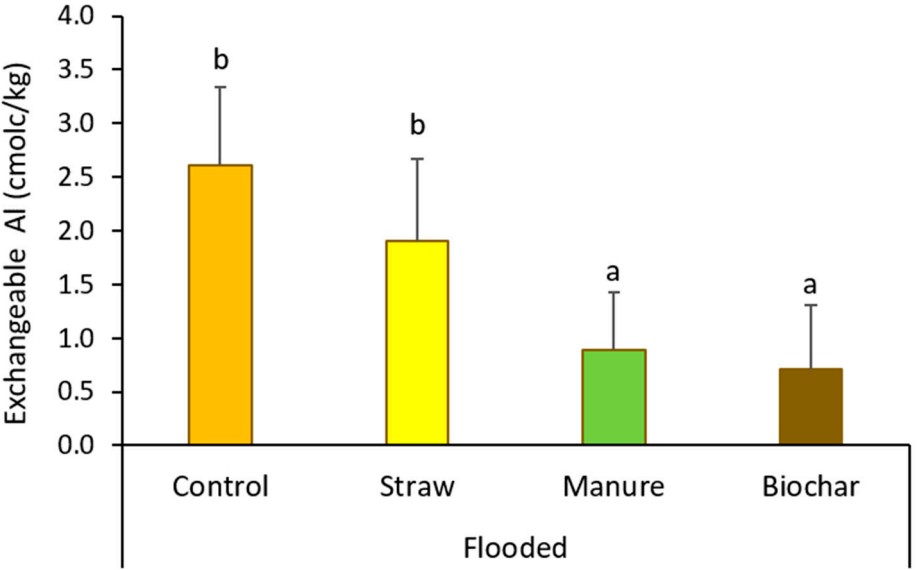

**Figure 10.** Exchangeable aluminium in soils after the second incubation experiment. Results correspond to the mean and standard deviation of six samples (*n* = 6). Statistically significant differences at *p* ≤ 0.05 due to amendments are indicated with lowercase letters.

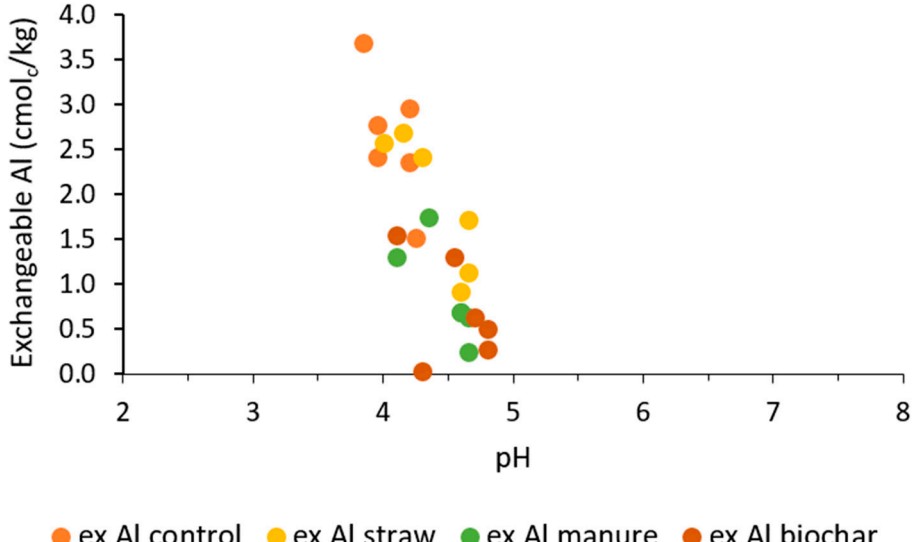

**Figure 11.** Relationship between soil pH and exchangeable aluminum in the soil at the end of the second incubation experiment.

*3.4. Soil Solution Composition after Incubation*

The addition of amendments changed the soil solution composition. As the initial salinity of the analysed soils exhibited wide variability, to detect trends in soil solution due to amendments, the concentrations in solution are expressed in relation to the soil solution composition in the control treatment for each soil sample (Table 3).

**Table 3.** Changes in the selected soluble elements' relative concentrations produced in ASS by adding organic amendments with anaerobic incubation (data are the mean $\pm$ standard deviation, $n = 6$; P = probability level of significant differences). Statistically significant differences at $p \leq 0.05$ due to amendments are indicated with lowercase letters.

| Amendment | Al/Alc | Ca/Cac | Fe/Fec | K/Kc | Mg/Mgc | Mn/Mnc | S/Sc |
|---|---|---|---|---|---|---|---|
| Control | 1 c | 1 a | 1 a | 1 | 1 a | 1 a | 1 ab |
| Rice straw | 0.09 $\pm$ 0.10 a | 0.55 $\pm$ 0.28 a | 714.77 $\pm$ 469.19 b | 3.19 $\pm$ 3.06 | 0.80 $\pm$ 0.27 a | 0.54 $\pm$ 0.38 a | 0.35 $\pm$ 0.39 a |
| Manure | 0.24 $\pm$ 0.15 b | 5.50 $\pm$ 1.92 b | 1.51 $\pm$ 1.14 a | 5.20 $\pm$ 4.84 | 3.02 $\pm$ 0.64 c | 2.19 $\pm$ 1.11 b | 3.68 $\pm$ 2.83 c |
| Biochar | 0.09 $\pm$ 0.06 a | 5.18 $\pm$ 1.91 b | 0.29 $\pm$ 0.16 a | 2.35 $\pm$ 1.77 | 1.51 $\pm$ 0.28 b | 2.35 $\pm$ 1.08 b | 2.29 $\pm$ 1.29 bc |
| P | 0.0000 | 0.0000 | 0.0000 | 0.1732 | 0.0000 | 0.0015 | 0.0075 |

The soil solution changed in response to the organic amendments and depended very much on the soil composition (Table 3). Although all the organic amendments reduced soluble Al, when comparing the effects on macronutrient content, the soils treated with rice straw significantly differed from the other two amendments, increasing soluble Fe but lowering Ca, K, Mg, and Mn. In addition, soluble sulphur was also lower than that in the control, which can be attributed to secondary sulphate reduction reactions that reduce sulphur solubility.

## 4. Discussion

Many marsh areas have been reclaimed for agriculture in the last 20 to 50 years by deforestation, drainage, and peat burning. The oxygenation of rich sulphur matter produces ASS that presents many plant-growing constraints. The most critical rice-growing constraints in ASS are aluminium toxicity and phosphorous deficiency, which are associated with low pH levels and Fe toxicity in field trials [12,13]. Tanaka and Navasero [50] also found a relationship with crop management, observing that the Al content of the soil solution diminished after flooding, and they concluded that the plant growth in the lowlands was not affected by aluminium toxicity, whereas the main cause of poor lowland rice growth in ASS was iron toxicity. Iron toxicity is associated with rapid drops in the redox potential upon soil flooding. High reducible iron levels combined with large amounts of organic matter hasten the severity of Fe toxicity symptoms. At the landscape level, most of the Fe-toxic lowlands in West Africa are located in humid forest zones with high rainfall.

The most significant effect of excess water is the isolation of soil from the atmosphere and the slowing of $O_2$ entering the soil. Once the soil reaches anaerobic conditions, other substances besides $O_2$ act as terminal acceptors of the free electrons produced during the respiration of soil microorganisms. In soil, redox reactions are coupled with different microorganisms whose metabolism develops when $O_2$ is absent, which involves distinct inorganic compounds as terminal acceptors for the free electrons that lower the soil redox potential as the free energy of the reduction reaction decreases. The first acceptor of electrons is nitrate, which is reduced to nitrogen oxides and, under more intense reducing conditions, $MnO_2$ or $Fe^{3+}$ can act as electron acceptors. $Fe^{3+}$ reduction begins a few days after inundation or at a redox potential of 180–150 mV [51]. Anaerobiosis and $Fe^{3+}$ reduction under submerged conditions lead to a rise in pH, which enhances reduction processes by organic matter that acts as an electron donor by lowering, in turn, the redox potential.

Different studies conducted under pot or field conditions have indicated that appropriate agronomic management increases rice production in ASS. To cope with iron toxicity,

several strategies have been proposed: add mineral fertilisers to improve soil fertility [38]; apply phosphogypsum and lime to increase available Ca and soil pH and lower toxic iron and aluminium levels [34,39,52]; add organic residue, which could stimulate the activity of reducing bacteria [32] to lead to a rise in pH under flooded conditions.

The effectiveness of organic matter amendments seems to be related to the composition of the applied residue [53]. The ameliorative effect of organic matter on preventing sulphidic soil oxidation depends, at least in part, on its decomposition [54]. Along these lines, Annisa et al. [55] found that adding fresh organic matter (e.g., rice straw) increased methane emissions, and thus, they recommend composting organic residue before applying it to the soil. Fresh rice straw consistently increases the $Fe^{2+}$ concentration in leachates [56]. Experiments in which compost or biochar has been added to ASS have reported higher rice yields [38,41,57]. During two consecutive field campaigns, Fall et al. [58] found the biochar amendment more effective than oyster shell in increasing rice production in the lower Casamance region.

During the first incubation experiment, different behaviour was observed when comparing the response to soil amendments under flooded conditions to free drainage, which could simulate lowland and upland rice production under field conditions. Under the flooding conditions, the pH of the soil solution in the control soils gradually rose, while the soil redox potential slightly lowered. Adding organic amendments resulted in greater pH increments, although the greatest pH increase came about with the lime amendment.

Considering that the redox potential decreases under 150 mV, the conditions for sulphate reduction and methanogenesis probably came about with the production of methane, ammonium, mercaptans, organic sulphides, and hydrogen sulphide through a large number of intermediary metabolites [59]. Our experiments showed that rice straw was the amendment that produced the highest soluble iron content and the lowest redox potential. Thus, in the initial flooding stages, the easily decomposable organic matter of rice soils is mineralised by microorganisms using $Fe^{3+}$ as an electron acceptor [60]. The amendments that produced the greatest reductions in exchangeable Al were lime in experiment 1 and biochar in experiment 2. The decrease in exchangeable aluminium following lime addition was caused by their precipitation as oxyhydroxides at the higher pH values found with the lime addition, whereas the reduction with biochar was accompanied only by a slight increase in pH. The organic macromolecules that form soil organic matter contain many functional groups, which play a major role in the sorption of ionic forms of trace elements. Organic matter apparently binds Al in an unexchangeable form.

The main toxicity symptom caused by Al in plants is root growth inhibition [5]. Toxicity is related to the monomeric Al concentration in solution, and the toxic effects of Al on roots are inhibited when pH increases due to either chelation with phosphate or organic acids or the reduction of monomeric Al species in solution [7]. Iron re-oxidation processes predominate over those of Fe reduction in the rhizosphere of most indica-type rice, which leads to considerable $Fe^{3+}$ accumulation and the formation of iron plaques around the rice roots [59]. Therefore, the rhizosphere of aerenchymal plants remains well-oxygenated and can support higher organic matter decomposition rates [59]. Increased tolerance to Fe toxicity can be associated with reduced Al in soil solution, which may increase root growth and the capacity to reduce Fe in soil solution by oxidation in the rhizosphere. This interaction favours plant growth at low pH values in soils with high organic matter content and in soils amended with stable organic matter.

In the second incubation experiment, in which changes were measured for the shortest times, different effects were found depending on the degree of organic matter stabilisation. The pH immediately rose after flooding with the addition of manure and biochar. By contrast, fresh organic matter (i.e., straw) brought about a greater reduction than stabilised organic matter. However, reduction peaked 5 days after flooding, as the responses mediated by microorganisms are slower than chemical responses. In other studies, microorganism activity increased with submergence duration and potentially reached peak values 2–8 weeks

after soil flooding [61]. Sulphate-reducing bacteria produce a variety of sulphur gases, including hydrogen sulphide, dimethylsulphide, and carbonyl sulphide [62,63].

All the organic amendments diminished the Al in the soil solution in experiment 2. However, whereas labile organic matter (i.e., straw) slightly decreased $Al_{ex}$, stabilised organic matter (i.e., manure and biochar) considerably reduced $Al_{ex}$ in the soil (as shown by both incubation experiments). This reveals a detoxing capability independent of pH, as both amendments produced only slight increments in pH compared with lime. Shetty et al. [64] found that wood biochar significantly decreased soluble and exchangeable Al, showing biochar's potential to decrease Al toxicity in acid soils. The biochar's adsorption and precipitation capacity are critical in Al toxicity alleviation compared with its alkalinity [65].

Rice straw degradation consumes oxygen, which results in negative redox values and an increase in the solution pH. In related research, Al diminished and Fe increased in the soil solution after wheat straw addition [66]. The increase in Fe in the soil solution and the production of organo-sulphur compounds indicate that straw, which is decomposable organic matter, lowers the soil redox potential to levels that induce Fe and sulphate reduction. The iron oxides on the inner container wall that formed during the experiment were probably the product of $Fe^{2+}$ oxidation. This $Fe^{2+}$ was generated by the reduction of Fe (III) in the soil, which requires organic carbon, and it was diffused to the overlaying water, where it was re-oxidised [52]. Consequently, the soil solution was rusty in colour and a fine orange layer appeared in the soil several mm under the soil surface. In a later work, Kölbl et al. [67] showed that under moderate reducing conditions, iron evolved into mineral forms (goethite and lepidocrocite), which were not prone to oxidation and re-acidification with future aeration.

The soil amendment applied to obtain a greater rice yield implies a rise in pH that produces a reduction in aluminium toxicity and an increase in nutrient availability. Lime addition increases pH, but not nutrient availability, except for those nutrients whose concentration is controlled by pH, like P. By using organic amendments, the strategy is to employ the reductant capacity potentiated by water management and Fe III is reduced by consuming protons, which leads to a higher pH. However, ferrous forms are more soluble, leading to iron toxicity risk. The reducing capacity and thus the $Fe^{2+}$ concentration in the soil solution are controlled by (1) the degree of oxygen arriving in the soil; (2) the original $Fe^{3+}$ concentration; (3) organic matter degradability, which is greater for fresh organic matter (straw) and lower for stabler organic matter (manure).

Although water management is crucial [68], in flood plains with rainfed agriculture, it is determined by rain distribution, and it is not easy to control water drainage whenever desired. In rainfed agriculture, crop yields depend on rainfall distribution. In 2019 in Santack Valley, the rice failed because of delayed rainfall in relation to normal sowing data. Given the significant decreasing trend in annual rainfall for the 1922–2015 period in southern Senegal, along with an increasing trend in the annual reference evapotranspiration [25], appropriate adaptive strategies should be implemented to diminish the adverse influence of increasing aridity on rice productivity. Irrigation infrastructure is generally deficient across West Africa [69], where large efficient public irrigation schemes need to be implemented. Implementing intermittent irrigation in the Casamance basin may, however, entail technological challenges because this requires sufficient water availability during critical growth periods.

Wetland sediments are characterised by complex temporal and spatial gradients on dominant metabolic pathways and organic matter oxidation rates. Crop reduction is not the only environmental problem associated with ASS. Acid drainage water can solubilise toxic metals and transport pollution to lower areas [70]. Improving water management systems can increase yield but is it important to take into account that runoff management in ASS is accompanied by worse water quality. A way to improve chemical quality is to treat drainage water with an alkaline amendment. As lime is difficult to find, other natural products with an alkaline reaction, such as oyster shells, can be used to pave waterways.

## 5. Conclusions

From the results of this study and the literature review, several aspects can be identified as key to improving rice production subjected to ASS constraints. The soil's capacity to retain and immobilise contaminants is influenced not only by the organic matter content but also by its complexity and degree of polymerisation, with the most recalcitrant compounds (biochar) showing maximum Al adsorption capacity and younger plant residue presenting minimum adsorption capacity.

A rapid increase in Fe (II) following flooding is favoured by a low initial soil pH, sustained organic matter supply, the presence of easily reducible Fe, a good soil fertility status, and the lack of compounds with a higher oxidation state than Fe(III) oxide, especially oxygen, Mn(III, IV) oxide, and nitrate, in the soil. In soils rich in reducible Fe and decomposable organic matter, soil reduction mobilises large amounts of water-soluble Fe, which is the apparent cause of Fe toxicity for rice. In this way, soil crops and soil response to amendments in ASS depend on soil characteristics, their oxidised compound content, and the quantity and quality of organic and mineral amendments, biochar being a potential amendment to alleviate Fe and Al toxicity in ASS. More research efforts under field conditions are necessary to control all the interacting factors that contribute to soil fertility in ASS.

**Supplementary Materials:** The following supporting information can be downloaded at: https://www.mdpi.com/article/10.3390/land12091693/s1, Table S1: Soil percolate properties in the free drainage and flooded treatment.; Table S2. Soil properties in the flooded treatments during the second incubation at different times after adding amendments.

**Author Contributions:** Conceptualisation, I.B., N.S. and J.M.O.; methodology, I.B., N.S., J.M.O. and J.O.; formal analysis, I.B.; investigation, I.B. and J.O.; resources, I.B., A.L., N.S. and J.M.O.; data curation, I.B.; writing—original draft preparation, I.B.; writing—review and editing, I.B., A.L. and N.S.; visualisation, I.B.; supervision, I.B.; project administration, I.B. and J.O.; funding acquisition, I.B., A.L., N.S. and J.M.O. All authors have read and agreed to the published version of the manuscript.

**Funding:** The authors acknowledge the Centre for Development Cooperation of the Universitat Politècnica de València (CCD-UPV) for providing funds as part of Project AD1810-UPV.

**Data Availability Statement:** Not applicable.

**Acknowledgments:** The authors thank the support of Caritas Spain (especially Soledad Gutiérrez and Pablo Reyero) and Caritas Senegal (especially Ludovic Seydou Diedhiou) for facilitating its implementation and the University Assane Seck of Ziguinchor (especially Sire Diedhiou for providing information about iron toxicity in Casamance soils).

**Conflicts of Interest:** The authors declare no conflict of interest.

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
