# Peer review of "Improving the Chemical Properties of Acid Sulphate Soils from the Casamance River Basin"

_land, doi:10.3390/land12091693_

Round 1
Reviewer 1 Report
In general the manuscript is written and organized well. Below are my comments on this manuscript should the authors take care off them:
1- The authors should add more numerical results in the Abstract.
2- Th important of study and goal should be clarified in the Abstract
3- More literatures about pH level should be presented.
4- Line 142: What is the reason of perform this study? This is should be briefly explained
5- Figure 1: Missing some important information such as the direction of the north and the map legend
6- The plots ( P1, P2, …) should be identified in the legend of Figure 1.
7- The soil characteristics along side with Table 1 should be moved to the method and material section.
8- More details about the difference between flooded and non-flooded treatments.
9- More physical observations about for Figures 7 and 8 should be provided and discussed.
10- Limitations of the study and future research should be provided in the Conclusion section.
Minor editing of English language required
Author Response
Black, italics and underlined: editor and reviewers’ comments.
Black: authors’ comments.
Blue: new text introduced into the manuscript.
REVIEWER 1
In general, the manuscript is written and organized well. Below are my comments on this manuscript should the authors take care off them:
Thank you for highlighting that the manuscript is written and organized well and for your valuable suggestions.
R1C2 The authors should add more numerical results in the Abstract.
Numerical results have been added in the abstract. It is necessary to take into account that the abstract is limited to 200 words.
R1C3 Th important of study and goal should be clarified in the Abstract
The goal of the study has been specified in the Abstract.
R1C4 More literatures about pH level should be presented.
New information on acid soils problems has been added:
Aluminium toxicity is a critical growth-limiting factor for plants in many acid soils, mainly affecting root growth, especially when soil pH values are less than 5.0 [5]. The toxicity to the plant roots is determined by the availability of monomeric species of Al [6, 7]. Phytotoxicity loss occurs when monomeric Al is diminished by the polymerization of Al by increasing pH [7, 8], forming complex non-exchangeable polymeric hydroxy-aluminum ions [9].
Bache B.W., Sharp G.S., Soluble polymeric hydroxy-aluminium ions in acid soils, J. Soil Sci. 27 (1976) 167–174.
Baes C.F. Jr., Mesmer R.E., The hydrolysis of cations, Wiley Interscience, New York, 1976, pp. 112–118.
Bartlett R.J., Riego D.C., Effect of chelation on the toxicity of aluminium, Plant and Soil 37 (1972) 419–423.
R1C5 Line 142: What is the reason of perform this study? This is should be briefly explained.
This study aims to compare the effect of different amendments on ASS remediation.
.
R1C6 Figure 1: Missing some important information such as the direction of the north and the map legend.
Figure 1 has been improved. The caption of Figure 1 has been changed as follows;
Figure 1. Casamance River Basin (a); location of the Sanctack Valley (b); and plots location into the Santack Valley (c).
R1C7 The plots ( P1, P2, …) should be identified in the legend of Figure 1.
The plots are now identified in the Figure 1.
:
R1C8 The soil characteristics along side with Table 1 should be moved to the method and material section.
Following the reviewer’s comment, the soil charateristics and Table 1 have been moved.
R1C9 More details about the difference between flooded and non-flooded treatments.
Additional information has been added to the material and methods section:
The percolation tubes are hand-made of glass; thus, they are not uniform. The internal diameter and the height were around 3.4 and 12 cm, respectively. The soil depth inside the percolation tube is around 5 cm. In the tubes corresponding to the flooded treatments, where drainage was not allowed, 40 mL of distilled water was added. In this way, in the flooded treatments, the soils had a constant water sheet of around 4.5 cm height. In the free drainage treatment, water exit was left open, in this way the soils are kept at field capacity humidity.
R1C10 More physical observations about for Figures 7 and 8 should be provided and discussed.
New observations have been added:
Adding straw reduced Alex to 68% and 84% of the control soil values in the flooded and non-flooded treatments, respectively. Manure addition reduced Alex to 45% and 44% of the control soil values in the flooded and non-flooded treatments, respectively. Alex values were lower under flooded conditions.
R1C11 Limitations of the study and future research should be provided in the Conclusion section.
Future research must focus on biochar amendments in ASS.
biochar being a potential amendment to alleviate Fe and Al toxicity in ASS.

Reviewer 2 Report
The introduction is correct, and the objectives are clearly established. I would suggest including some data on the biogeochemistry of aluminium because of the importance of pH and because the third objective is related to analysing the effect of soil pH and exchangeable aluminium.
I have detected some typographical errors. Not many. Line 62, NO3- and line 147 change Fe III to Fe (III)Material and Methods are also clear and robust; however, the authors should provide some extra information. For example, columns should be detailed (material, length and diameter, also soil depth during the experiment). Also, the organic matter added with the amendments should be specified in line 200. Please, see the comments below:
Line 176: give more information about the plots and the plateau (extension, geographical situation, number of samples taken to make the composite sample, depth in the soil where the samples were taken…). Did all the plots have the same productivity? I would suggest providing this information in the description of the experimental design.
Line 177: Indicate which diameter fractions were analysed, specify if samples were ground and how samples were dried.
Line 219: Please, indicate that soluble Fe was determined in the aqueous fraction (acidified?). It is unclear how the samples were taken.
Line 194: First incubation experiment. Give some information about where the amendment came from. And how they were characterized. Indicate the number of replicates
Line 223: Change Alex to interchangeable Al
Line 227-243: Second incubation experiment: no number of replicates is given.
Results section
Table 2: I consider that Fe, Al and S contained in Biochar should be included in the table.
Figures 2,3, 5, 6, and 7: Indicate the meaning of the error bars, the number of samples and whether the statistical differences are significant according to Duncan's test.
Figure 5: The lowercase letters are missing. If there are no differences, please remove them.
The discussion, conclusion and references are correct.
Please, review English grammar, v.g.: lines 131-133 say “Compared to complex organic matter, the addition of single compounds, such as glucose or nitrogen salts, results in a slight pH increase [28]. Addition of organic amendments has also been demonstrated to increase rice production, although its efficacy also depends on the type of the organic amendment applied.” It should read: “Compared to complex organic matter, the addition of single compounds, such as glucose or nitrogen salts, results in a slight pH increase [28]. The addition of organic amendments has also been demonstrated to increase rice production, although its efficacy also depends on the type of organic amendment applied.”
Author Response
REVIEWER 2
R2C1 The introduction is correct, and the objectives are clearly established. I would suggest including some data on the biogeochemistry of aluminium because of the importance of pH and because the third objective is related to analysing the effect of soil pH and exchangeable aluminium.
Resp:
Thank you, We agree with the reviewer. We have added additional information about aluminium chemistry both in the introduction and discussion sections.
Introduction
Aluminium toxicity is a critical growth-limiting factor for plants in many acid soils, mainly affecting root growth, especially when soil pH values are less than 5.0 [5]. The toxicity to the plant roots is determined by the availability of monomeric species of Al [6, 7]. Phytotoxicity loss occurs when monomeric Al is diminished by the polymerization of Al by increasing pH [7, 8], forming complex non-exchangeable polymeric hydroxy-aluminum ions [9].
Bache B.W., Sharp G.S., Soluble polymeric hydroxy-aluminium ions in acid soils, J. Soil Sci. 27 (1976) 167–174.
Baes C.F. Jr., Mesmer R.E., The hydrolysis of cations, Wiley Interscience, New York, 1976, pp. 112–118.
Bartlett R.J., Riego D.C., Effect of chelation on the toxicity of aluminium, Plant and Soil 37 (1972) 419–423.
Discussion
Shetty et al. [64] found that wood biochar significantly decreased soluble and ex-changeable Al, showing the biochar potential to decrease the Al toxicity in acid soils. The biochar´s adsorption and precipitation capacity are critical in Al toxicity alleviation compared with its alkalinity [65].
R2C2 I have detected some typographical errors. Not many. Line 62, NO3- and line 147 change Fe III to Fe (III).
Thank you, typographical errors have been corrected.
R2C3 Material and Methods are also clear and robust; however, the authors should provide some extra information. For example, columns should be detailed (material, length and diameter, also soil depth during the experiment). Also, the organic matter added with the amendments should be specified in line 200. Please, see the comments below:
Thank you for your suggestion. Extra information on the laboratory experiment has been added.
The percolation tubes are hand-made of glass; thus, they are not uniform. The internal diameter and the height were around 3.4 and 12 cm, respectively. The soil depth inside the percolation tube is around 5 cm. In the tubes corresponding to the flooded treatments, where drainage was not allowed, 40 mL of distilled water was added. In this way, in the flooded treatments, the soils had a constant water sheet of around 4.5 cm height. In the free drainage treatment, water exit was left open, in this way the soils are kept at field capacity humidity.
Organic matter addition, 4.8 and 2.8 g C kg-1 soil added with straw and manure respectively, was similar to the values of oxidable organic carbon initially found in the soils,
R2C4 Line 176: give more information about the plots and the plateau (extension, geographical situation, number of samples taken to make the composite sample, depth in the soil where the samples were taken…). Did all the plots have the same productivity? I would suggest providing this information in the description of the experimental design.
Thank you for your suggestion. We added additional information.
In July 2019, four agricultural plots were selected, advised by the technique staff from Caritas Ziguinchor. Plots (labelled as P1, P2, P3 and P4) were selected for their productivity differences fromin the previous campaign. Specifically, plot P1 had null productivity in the last campaign and the remaining plots located in the highest level of the valley were more productive. To characterise soil properties, a composite soil sample was obtained in each plot, made of 4 samples taken from 0 to 10 cm. A single soil sample was taken in the plateau near the valley from 0 to 10 cm. Soil samples were transported to the laboratory, air dried, ground and sieved through a 2 mm mesh.
R2C5 Line 177: Indicate which diameter fractions were analysed, specify if samples were ground and how samples were dried.
We hope that with the clarification made following R2C4 this concept is clear.
R2C6 Line 219: Please, indicate that soluble Fe was determined in the aqueous fraction (acidified?). It is unclear how the samples were taken.
Soluble Fe was determined in aqueous percolates without acidification. This information was added.
R2C7 Line 194: First incubation experiment. Give some information about where the amendment came from. And how they were characterized. Indicate the number of replicates.
Thank you. The following information was added:
Rice straw was obtained from a rice production zone in L’Albufera (Valencia, Spain) in September 2019. The sample was air dried, ground and sieved through a 0.5 mm mesh size. The manure is sheep manure obtained from an organic agricultural plot in Pedralba (Valencia, Spain). Manure was air dried, ground and sieved through a 0.5 mm mesh size.
The biochar was purchased from Piroeco Bioenergy S.L. (Malaga, Spain). It was produced from holm oak by slow pyrolysis at 650â—¦C and atmospheric pressure, the residence time in the reactor chamber being 12–18 h. Biochar was air dried, ground and sieved trough a 0,5 mm mesh. The elemental composition of the biochar was obtained from Saez et al [48].
R2C8 Line 223: Change Alex to interchangeable Al
Thank you. It has been changed.
R2C9 Line 227-243: Second incubation experiment: no number of replicates is given.
The six soils are considered as replicates. Number of replicates = 6. This information has been added to figure captions.
R2 C10 Results section
Table 2: I consider that Fe, Al and S contained in Biochar should be included in the table.
Fe and S in biochar has been included. Al was not determined.
R2C11 Figures 2,3, 5, 6, and 7: Indicate the meaning of the error bars, the number of samples and whether the statistical differences are significant according to Duncan's test.
Thank you for the suggestions, now the meaning of the error bars and the number of samples are included in the figure captions.
R2C12 Figure 5: The lowercase letters are missing. If there are no differences, please remove them.
Thank you very much. It was an error. The figure was changed, and lowercase letters incorporated.
R2C13 The discussion, conclusion and references are correct.
Thank you for your comments and for your valuable suggestions.
.
R2C14 Please, review English grammar, v.g.: lines 131-133 say “Compared to complex organic matter, the addition of single compounds, such as glucose or nitrogen salts, results in a slight pH increase [28]. Addition of organic amendments has also been demonstrated to increase rice production, although its efficacy also depends on the type of the organic amendment applied.” It should read: “Compared to complex organic matter, the addition of single compounds, such as glucose or nitrogen salts, results in a slight pH increase [28]. The addition of organic amendments has also been demonstrated to increase rice production, although its efficacy also depends on the type of organic amendment applied.”
Thank you. The change has been done and English has been revised.
Please see the attachment

Round 2
Reviewer 1 Report
The authors did an excellent job of answering my concerns. It is good to go
Author Response
Thank you for your review.
Reviewer 2 Report
All of the above comments have been addressed to a greater or lesser degree.
A single recommendation is to include the number of replicates for each type of the 6 different soils considered in this study in lines 273-296 that correspond to section "2.3. Second incubation experiment. Second incubation experiment". Supposedly estimated mean and standard deviation for four samples (n=4). Please include this information in this section 2.3.
Author Response
Thank you for your review.
Following the reviewer suggestions, new information has been added.
lines 262-263: A total of 32 incubation tubes were prepared ( 4 soils x 4 amendment treatments x 2 water management treatments).
lines 283-285: The incubations were performed for each combination of soil and treatment, considering the soils as replications of the amendment treatments.